# Prolonged Grief Symptoms among Suicide-Loss Survivors: The Contribution of Intrapersonal and Interpersonal Characteristics

**DOI:** 10.3390/ijerph191710545

**Published:** 2022-08-24

**Authors:** Yossi Levi-Belz, Tamir Ben-Yaish

**Affiliations:** 1The Lior Tsfaty Center for Suicide and Mental Pain Studies, Ruppin Academic Center, Emek Hefer 40250, Israel; 2Clinical Psychology of Adulthood and Aging M.A. Program, Ruppin Academic Center, Emek Hefer 40250, Israel

**Keywords:** suicide, bereavement, prolonged grief, guilt, self-disclosure, belongingness

## Abstract

Background: Suicide-loss survivors (SLSs) are a population with unique characteristics that place them at increased risk for developing grief complications and painful feelings of guilt that may impact their supportive social environment. However, no studies to date have examined the role of intrapersonal and interpersonal variables that may contribute to prolonged grief symptoms (PGS) as outlined by the new DSM-5 criteria. The present study aimed to extend knowledge regarding the role of interpersonal variables, such as perceived burdensomeness, thwarted belongingness, and self-disclosure, in determining the impact of guilt on the development of PGS among SLSs. Method: This study is part of a longitudinal study, though, in this study, we used a cross-sectional examination of the recently completed fourth measurement. Study participants included 152 SLSs aged 22 to 76 who completed questionnaires measuring guilt, depression, perceived burdensomeness, thwarted belongingness, self-disclosure, and PGS using the Prolonged Grief–Revised Inventory. Participants’ demographics and loss-related characteristics, such as time since suicide and participant’s age at the time of suicide, were examined. Results: Confirming the hypotheses, intrapersonal variables (i.e., guilt and depression), as well as interpersonal variables (i.e., perceived burdensomeness, thwarted belongingness, and self-disclosure), contributed significantly to PGS beyond sociodemographic and loss-related factors. Perceived burdensomeness significantly moderated the contribution of guilt to PGS: for participants with high burdensomeness levels, guilt contributed to PGS more strongly than for participants with low burdensomeness. Conclusion: Guilt is an important contributor to PGS among SLSs, and perceived burdensomeness plays a critical role in moderating this contribution. In light of these findings, it can be suggested that SLSs with high levels of guilt should receive special attention and may benefit from therapeutic interventions focusing on reducing maladaptive cognitions that elicit intense guilt or perceived burden.

## 1. Introduction

### 1.1. Prolonged Grief Disorder (PGD)

Grief is a universal, inevitable, painful part of life that has its origins in the severing of close relationships [1]. The psychological response to the loss of a loved one is usually accompanied by grief and emotional distress. Most bereaved individuals go through a natural grieving process in which the intensity of symptoms decreases over time as the bereaved gradually acclimates to the impact of the loss [1]. However, a significant minority of these individuals experience abnormally intense and persistent grief symptoms, resulting in significant functional impairment [2]. For decades, these and other symptoms following loss were equated with symptoms of depression. Recently, cumulative evidence has indicated that prolonged grief symptoms (PGS) represent a bereavement-specific diagnosis [3] that comprises part of the clinical syndrome of prolonged grief disorder, a condition recently supplemented to the International Classification of Diseases, 11th Revision (ICD-11; [4]). Prolonged grief disorder is now included for the first time in the Diagnostic and Statistical Manual of Mental Disorders, Fifth Edition, Text Revision (DSM-5-TR) in the chapter on trauma and stressor-related disorders [3,5].

Although PGD is an identifiable and distinct disorder [6], research suggests it should be viewed as a continuum of normal to prolonged and severe grief reactions rather than a qualitative category [6,7]. Therefore, clinically relevant levels of prolonged grief symptoms (PGS) are commonly used as a proxy to identify prolonged grief disorder [8].

The core element of PGD is intense longing or craving for the deceased or engaging in thoughts or memories of them [3]. Other symptoms include intense emotional pain related to the death, feeling that a part of oneself has been lost, difficulty accepting the death, avoiding cues that might trigger thoughts of the deceased, and difficulty engaging in social or other activities following the loss [3]. According to DSM-5 criteria, symptoms should persist for at least 12 months after the loss [9], whereas ICD diagnostic criteria anticipate symptoms persisting for more than six months after the loss [5].

Until the criteria for PGD were established and recognized, researchers and clinicians used other related concepts, such as complicated grief, pathological grief, persistent complex bereavement disorder, and traumatic grief [10]. These constructs were assessed in recent decades by various standards, instruments, and cutoff scores for disordered grief, such as PG-13, ICD-11 criteria, and ICG-R [11]. These PG-related structures have been associated with various adverse health outcomes [1], substance abuse [12], increased suicide risk [13], and decreased psychological quality of life [14]. However, in the absence of valid self-report measures that met the DSM-5-TR criteria of PGD [3], these findings could not be generalized to enable a fuller understanding of the prevalence of PGD and its correlates. A population that should be assessed using the new DSM-5-TR criteria is suicide-loss survivors, who have recently been classified among those at highest risk for developing PGS [15].

### 1.2. Suicide-Loss Survivors

Suicide is a significant public health problem, causing an estimated 700,000 deaths annually, making it the seventeenth leading cause of death worldwide [9]. Each death caused by suicide can affect approximately 135 people [16]; these are referred to as suicide-loss survivors (SLSs). Many SLSs are at high risk of experiencing severe emotional, physiological, or social distress over an extended period following the suicide of a significant other [17,18,19]. Moreover, SLSs are particularly at risk for developing PGS [15], which the particular nature of suicide grief can explain. Most SLSs have difficulty understanding the cause of death and why their loved one chose to end their life [20]. They may be dominated by the grieving process, resulting in higher levels of PGS [21,22]. Therefore, it is essential to understand the psychological contributors and moderators of PGS among SLSs.

### 1.3. Intrapersonal and Interpersonal Factors

It is well-established in the literature that SLSs often suffer from feelings of shame, rejection, and social stigma [17,23]. However, one factor that has been repeatedly highlighted, and is perhaps the most common reaction in suicide grief, is guilt [24,25,26]. Guilt is a remorseful emotional reaction, a sense of failure in one’s relationship with the deceased and of not living up to one’s standards and expectations [27]. Guilt following a suicide loss often emerges from the belief that the suicide could have been avoided had the bereaved behaved differently, such as being more attentive [1]. As a result, many SLSs are disinclined to share their thoughts and feelings with others [28,29]. This aversion to sharing can undermine the foundation of SLSs’ interpersonal relationships [18], promote loneliness and withdrawal, and exacerbate adverse psychological outcomes such as depression, anxiety [1], and hopelessness [30]. Recently, Feigelman and Cerel [25] showed that feelings of blameworthiness (a factor closely related to guilt) are associated with grief difficulties among bereaved parents of children who died by suicide. However, despite this widespread understanding of the importance of guilt in the suicide grieving process, few studies have focused on examining the factors that may facilitate or exacerbate its contribution to the development of PGS. Depression also characterizes SLSs, as it reflects the high mental burden that SLSs may carry in the aftermath of suicide loss [17]. Several studies have found that suicide bereavement is related to higher levels of depression than among other bereaved individuals [31]. Considering that depression is closely related to PGS in general and among SLSs in particular, it is crucial to investigate depression’s role alongside the guilt experience. 

Several studies have suggested that interpersonal factors are among the main contributors to distress, depression, and complicated grief reactions among SLSs (e.g., [29,32,33]). The widely recognized interpersonal theory of suicide (ITS; [34]) posits that two interpersonal constructs––perceived burdensomeness and thwarted belongingness—are the primary causal interactional risk factors for distress and suicidality [35]. Each dimension represents a particular perception of the interpersonal environment, viewed as a dynamic cognitive–affective state. Perceived burdensomeness (PB) refers to an individual’s sense that their existence is a burden to friends, family members, or society and consists of two facets: 1. the belief that oneself is so flawed that one is a burden to others and 2. affectively charged cognitions of self-hatred [36]. Thwarted belongingness (TB) is the experience of loneliness and the lack of relationships based on reciprocity [36]. The sense of not belonging highlights the painful feeling of being external to the family, friends, and other social groups [36]. 

Both TB and PB were found to be related to several adverse outcomes in the aftermath of suicide loss. For example, recent research found that TB and PB predicted the extent of complicated grief over time in a sample of SLSs [22], while the experience of high belongingness promoted the development of higher levels of adaptive functioning after suicide loss [37]. We can thus assume that higher levels of PB and TB could contribute to higher levels of PGS among SLSs. Guilt is one of the painful feelings reported in post-suicide grief [25], and it may have intrapersonal and interpersonal ramifications that can undermine SLSs’ interpersonal relationships [28]. Indeed, it is likely that interpersonal perceptions such as TB and PB play a critical role in PGS among SLSs who suffer from guilt. 

Recent studies have shown that self-disclosure (SD)—a process by which people communicate themselves to others [38]—can help reduce distress and suicidal ideation [39] during the adjustment of SLSs [29] and even lead to posttraumatic growth [32]. SD promotes the healing process and reduces emotional distress by providing new perspectives on the self and the suicide event and helping to create constructive narratives about the changes that have occurred [40,41]. SD also provides intimacy, a sense of being with others, and an experience of social support [42] that allows one to overcome the walls of stigma and distance [43]. As SLSs tend to experience painful guilt [25], disclosing intimate feelings and thoughts is likely to be difficult for them, evoking feelings of ambivalence and psychological pain they may prefer to avoid [28]. A limited level of SD is liable to comprise a buffer between the individual and their social environment, leading to loneliness and social isolation and exacerbating the difficulties of bereavement [43,44]. Interestingly, no studies to date have examined the interpersonal contributors to PGS following suicide loss.

### 1.4. The Present Study

Considering the formal recognition of prolonged grief disorder as a new diagnostic entity and the evidence that it is associated with various health problems and impairments in quality of life, there is an increasing need to identify the factors contributing to the severity of PGS. However, few studies have examined such predictors of PGS among SLSs. To address this gap, this study aimed to examine the role of intrapersonal characteristics (i.e., guilt and depression) and interpersonal characteristics (i.e., TB, PB, and SD) and the impact of their co-occurrence as potential factors contributing to PGS after a suicide loss, using the revised DSM-TR diagnosis of PGD. In light of the linkage between depression and PGS, we also aimed to examine the contribution of guilt to PGS beyond the contribution of depression. Overall, identifying factors contributing to the development of PGS will facilitate the design of preventive interventions for individuals at increased risk and establish therapeutic measures targeting elements contributing to the onset and severity of PGS.

Three hypotheses were at the core of this study:

**Hypothesis** **1.**
*Higher levels of guilt will contribute to higher levels of PGS among SLSs, beyond sociodemographic and loss-related factors (i.e., the time elapsed since the loss, participant’s age at the time of suicide, and fear of a future suicide of a family member) and beyond depression levels.*


**Hypothesis** **2.**
*Lower levels of self-disclosure and higher levels of perceived burdensomeness and thwarted belongingness will contribute to higher levels of PGS among SLSs, beyond sociodemographic, loss-related, and intrapersonal factors (guilt and depression).*


**Hypothesis** **3.**
*Interpersonal factors (TB, PB, and SD) will moderate the relationship between guilt and PGS among SLSs. As SD scores decrease and TB and PB increase, the relationships between guilt and PGS will become stronger.*


## 2. Method

### 2.1. Participants

The present study is part of a longitudinal study with four measurement points (see [22]). The present study used the fourth measurement point, conducted in 2020–2021. A total of 189 SLSs participated in the first measurement point of the study, conducted in 2015–2016. Of these, 152 (80.4%) participated in the fourth measurement. Thus, the final sample of the current study comprised 152 suicide survivors (80.4% women), ranging in age from 22 to 76.

Participants were recruited in two main ways. Most participants were recruited through a nonprofit organization, The Path to Life, the national agency for suicide survivors in Israel. The remaining participants were recruited through the official Facebook group of Israeli suicide survivors and other social media groups of suicide survivors in Israel. Inclusion criteria for suicide survivors were individuals who had lost a family member or other close friend due to suicide. Exclusion criteria were age less than 15 years at the time of suicide and inability to read and write Hebrew.

### 2.2. Measures

#### 2.2.1. Guilt

Guilt levels were measured with Trauma-related guilt (TRG; [45]), an abbreviated form of the Trauma-Related Guilt Inventory (TRGI; [46]). The inventory assesses guilt characteristics following a traumatic event with five items representing the severity of guilt related to the experienced trauma (e.g., “I did something I should not have done”, reflecting a negative evaluation of the self). All items were measured on a Likert-type scale, ranging from 0 (Does not match at all) to 10 (Matches completely), resulting in a mean score of 0–10. Øktedalen et al. [45] reported the scale’s high reliability and content validity. For the present sample, Cronbach’s alpha was α = 0.87.

#### 2.2.2. Depression

Depression symptoms were assessed using the 9-item Patient Health Questionnaire Depression Scale (PHQ-9; [47]). Participants were asked, “Over the last two weeks, how often have you been disturbed by any of the following problems?” They then rated the frequency of each of the symptoms (e.g., Poor appetite or overeating? Little interest or pleasure in doing things?) using the following four anchors: 0 (not at all), 1 (several days), 2 (more than half of the days), and 3 (nearly every day). For this study, the mean of the scale scores was used, ranging from 0 to 27. Higher scores indicated more severe depressive symptoms. High PHQ-9 scores have been associated with increased physician visits, physical disability, risk of psychiatric comorbidity, and overall severity of the syndrome [47]. Cronbach’s alpha for the present sample was α = 0.90.

#### 2.2.3. Thwarted Belongingness and Perceived Burdensomeness

Thwarted belongingness and perceived burdensomeness were assessed using the 10-statement Interpersonal Needs Questionnaire (INQ; [35]). Five items evaluated each of the two subscales: thwarted belongingness (e.g., “These days, other people care about me” [reversed scored]) and perceived burdensomeness (e.g., “These days, I feel like a burden on the people in my life”). Respondents indicated the extent to which the statements applied to them on a 7-point Likert-type scale, with higher scores reflecting more severe levels of thwarted belongingness and perceived burdensomeness. In this study, we used the Hebrew translation of the INQ, which has been employed in several studies (e.g., [22]). Cronbach’s alpha for the present sample was α = 0.85 for thwarted belongingness and α = 0.90 for perceived burdensomeness.

#### 2.2.4. Self-Disclosure

The Distress Disclosure Index (DDI; [48]) was used to measure the individual’s inclination to disclose personally distressing information (e.g., “I usually don’t share issues that bother me” [reverse scored]; “I try to find people to talk to about my problems”). The 12-item DDI is presented on a 5-point Likert-type scale, ranging from 1 (strongly disagree) to 5 (strongly agree). Higher scores represent a greater inclination to disclose. Confirmatory factor analysis of the DDI yielded a single construct with high reliability and validity [48]. Cronbach’s alpha for the present sample was α = 0.94.

#### 2.2.5. Prolonged Grief Disorder (PGD) Symptoms

PGD symptoms were assessed using the Prolonged Grief-13-Revised (PG-13-R) scale [3]. This 10-item questionnaire is presented on a 5-point Likert-type scale, ranging from 1 (not at all) to 5 (overwhelmingly). The sum of the scale is used for the assessment of grief intensity on a dimensional scale for diagnosing PGD according to the new DSM-5-TR criteria (yearning, preoccupation, identity disruption, disbelief, avoidance, intense emotional pain, difficulty with reintegration, emotional numbness, feeling that life is meaningless, and intense loneliness). Higher scores reflect higher levels of PGS, with a PG-13-R symptom score of 30 or above indicative of syndrome-level PGD symptomatology [3]. 

The PG-13-R scale also presents three items, responded to dichotomously (“yes” or “no”), that are not included in the total score. They assess three diagnostic criteria: whether the respondent has lost a significant other, how long ago the death occurred, and the impairment associated with the symptoms. The PG-13-R scale has been shown to be reliable for measuring grief symptoms on a dimensional scale in three different community-based populations [3,49]. Cronbach’s alpha for the present sample was α = 0.87.

#### 2.2.6. Demographic and Loss-Related Characteristics

Demographic and diagnostic data on suicide loss were collected for each participant. These data included the age of the survivor and the deceased at the time of the suicide, the time elapsed since the suicide, the participant’s relationship to the person who died by suicide (e.g., child, spouse, parent, friend), the extent to which the suicide was expected or unexpected, and the extent of the fear of a future suicide of a family member.

### 2.3. Procedure

The study was approved by the ethics committee of the Ruppin Academic Center. Potential participants were informed of the risks and compensation procedures. They were assured of anonymity, confidentiality, and the right to withdraw from the study at any time. Participants were required to confirm their willingness to participate by signing an informed consent form and completing the questionnaire online (using Qualtrics online survey software). After completing the questionnaire, participants were asked if they consented to being contacted for a follow-up in the future. Consenting participants were invited to subsequent measurements (T2, T3, and T4) and were requested to complete a brief questionnaire that included the main study variables. After completing the online questionnaire, participants were compensated with vouchers at each measurement point (approximate value at each measurement: USD 25).

### 2.4. Statistical Analysis

Pearson correlation tests were calculated to examine the relationships among the study variables, followed by a hierarchical multiple regression with PGS as the dependent variable. As Aiken, West, and Reno [50] recommended, all continuous predictor variables were standardized, as were the cross-product interaction terms. To examine the nature of the interaction within a regression framework, moderation analysis was performed using the PROCESS macro (Model 1; [51]). The Statistical Package for the Social Sciences (SPSS, v26.0 for Windows, Armonk, NY, USA) was used for all analyses. A Bonferroni correction was applied to all analyses. The level of statistical significance was set at *p* = 0.05.

## 3. Results

### 3.1. The Sample’s Demographics

The mean age of the sample was 47.5 (*SD* = 14.7). Regarding participants’ marital status, 69 (45.4%) reported being married, 49 (32.2%) single, 20 (13.1%) divorced, and 14 (9.2%) widowed. Regarding religiosity, most of the participants (112, 73.7%) reported being secular, 29 (19.1%) reported being moderately religious, and the others (11, 7.3%) reported being religious Jewish. Regarding socioeconomic status (SES), 35 (23%) participants reported a very low SES, 38 (25%) low SES, and the remaining 79 (52%) medium and high SES. Regarding schooling, almost all participants (*n* = 132, 86.9%) reported completing at least 12 years of education, and 96 (63.2%) reported having a university degree.

### 3.2. Suicide-Related Demographic Information

The time elapsed since suicide varied, with a mean of 166 months (*SD* = 105.5). The mean age of participants at the time of suicide was 30.7 (*SD* = 10.1). Similarly, the mean age of the deceased at the time of suicide was 30.8 (*SD* = 13.9). Regarding the expectation of the suicide, only 9 (5.9%) participants indicated that the suicide was expected to a high degree, whereas 47 (30.9%) participants indicated that suicide was wholly unexpected. Regarding the nature of the participant’s relationship with the person who died by suicide, 26 (17.1%) participants reported losing a parent, 43 (28.3%) lost a sibling, 29 (19.1%) lost a child, 16 (10.5%) lost a spouse, 9 (5.9%) lost a relative (uncle, aunt, cousin), and 29 (19.1%) lost a close friend.

### 3.3. Intercorrelations of Study Variables 

Pearson correlations between the study variables were calculated for preliminary data analysis. The means, standard deviations, and intercorrelations are presented in Table 1. The matrix shows that PGS were significantly and positively correlated with the TB and PB dimensions of the interpersonal theory of suicide and negatively correlated with SD. In addition, a relatively strong positive correlation was found between PGS and depression and between PGS and guilt. Furthermore, scores for PGS were positively associated with participants’ age at the time of suicide and the fear of a future suicide by a family member. However, the time elapsed since the suicide was not significantly associated with PGS.

### 3.4. Hierarchical Regression Analyses

A four-step hierarchical multiple regression was performed to determine if interpersonal variables (PB, TB, and SD) predict the level of PGS among SLSs, beyond demographics, depression, and guilt variables [52] and with PGS as the dependent variable. In Step 1, three variables (time elapsed since the suicide, the age of the participant at the time of the suicide, and the level of fear of a future suicide of a family member) were entered into the equation. In Step 2, the intrapersonal factors (depression and guilt) were entered into the equation. In Step 3, the interpersonal characteristics (PB, TB, and SD) were entered into the equation to examine their contribution beyond the intrapersonal factors. In the final step, the Guilt X PB and the TB X SD interactions were added to the model.

Overall, the model was significant and explained 60% of the variance in PGS (*F*(11, 140) = 18.95, *p* < 0.001). Table 2 presents the contribution of each variable when entered into the regression. In Step 1, suicide-related demographic variables explained 24.6% of the variance (*F*(3, 148) = 16.09, *p* < 0.001). Specifically, we found that the participant’s age at the time of suicide had a significant positive contribution to PGS (β = 0.34, *t* (148) = 4.98, *p* <.001). Fear of future suicide by a family member also had a significant positive contribution to PGS (β = 0.32, *t* (148) = 5.30, *p* < 0.001). However, time since suicide did not significantly contribute to PGS. In Step 2, depression and guilt contributed by 24.4% to the explained variance beyond suicide-related demographic variables (*F*(2, 146) = 34.94, *p* < 0.001). Specifically, both depression (β = 0.45, *t* (146) = 6.10, *p* < 0.001) and guilt (β = 0.19, *t* (146) = 2.67, *p* < 0.01) had a significant positive contribution to PGS. In Step 3, PB, TB, and SD contributed an additional 6% of the variance beyond the intrapersonal and suicide-related demographic variables (*F*(3, 143) = 5.79, *p* < 0.001). Specifically, TB had a significant positive contribution to PGS (β = 0.32, *t* (143) = 3.80, *p* < 0.001). In addition, SD had a significant negative contribution to PGS (β = −0.21, *t* (143) = −3.07, *p* < 0.01). However, PB yielded only a marginally significant positive contribution to PGS. At the final step, the three interactions between guilt and PB, TB, and SD significantly contributed to PGS beyond all other variables, accounting for an additional 5.3% of the total variance (*F*(3, 140) = 6.15, *p* < 0.001). Specifically, the interaction of guilt and PB had a significant contribution to PGS (β = 0.46, *t* (140) = 3.87, *p* < 0.001). Additionally, the interaction of guilt and TB had a significant negative contribution to PGS (β = −0.30, *t* (140) = −2.42, *p* < 0.05). The Guilt X SD interaction did not significantly contribute to PGS.

#### Moderation Analyses

To understand the nature of the significant interactions, two moderation analyses were conducted using the PROCESS macro (Model 1; [51]). Two moderation analyses were performed with standard scores (Z) of PGS as the dependent variable and guilt as the independent variable. The moderators were PB and TB levels.

As presented in Figure 1, a significant interaction was found between guilt and PB in predicting PGS (*b* = 0.11, *SE* = 0.04, 95% CI [0.03, 0.17], *t* (172) = 2.83, *p* < 0.01). Examining the interaction revealed that for SLSs with higher levels of PB, guilt had a positive and stronger contribution to PGS (for low PB: *b* = 0.34, *SE* = 0.09, 95% CI [0.16, 0.52], *t* (172) = 3.81, *p* < 0.001; for moderate PB: *b* = 0.44, *SE* = 0.07, 95% CI [0.31, 0.58], *t* (172) = 6.38, *p* < 0.001; for high PB: *b* = 0.55, *SE* = 0.07, 95% CI [0.42, 0.68], *t* (172) = 8.12, *p* < 0.001). The interaction between guilt and TB in predicting PGS was not found to be significant in the moderation analysis (*b* = 0.03, *SE* = 0.03, 95% CI [−0.03, 0.09], *t* (172) = 0.98, *p* = 0.328).

## 4. Discussion

To our knowledge, this study is the first to demonstrate the critical contribution of interpersonal variables to the relationship between guilt and PGS in the aftermath of suicide loss by using the revised DSM−TR diagnosis of PGD. As SLSs are at increased risk of experiencing guilt, shame, and stigma [25], it is critical to understand the mechanisms that facilitate (or buffer) the contribution of such feelings to the development of PGS.

Our results indicated that guilt and depression are critical contributors to the level of PGS beyond sociodemographic and loss-related factors. These findings align with several studies that have highlighted the importance of guilt as a significant predictor of the development or progression of psychopathological symptoms in grief following suicide loss [19,30] and grief in general [53,54,55]. Other studies have suggested that feelings of guilt are longer lasting and more intense in suicide grief than in other forms of grief [56,57].

Furthermore, we found that interpersonal variables (i.e., perceived burdensomeness, thwarted belongingness, and self-disclosure) contributed to PGS beyond intrapersonal, sociodemographic, and loss-related factors. Given the strong association between depression and PGS, any contributions beyond depression can be considered highly noteworthy. Thus, our results suggest that the SLSs’ devastating feelings of lack of belonging, isolation, being a burden to their surroundings, and difficulty disclosing intimate feelings and thoughts may contribute to PGS. This finding aligns with several cross-sectional and longitudinal studies that highlight the critical role of these variables in maladaptive grief response [22,30,58,59].

This study’s main finding was that PB levels significantly moderated the contribution of guilt to PGS. Thus, it can be inferred from these results that perceived burdensomeness plays a significant role in facilitating PGS among SLSs, especially those experiencing guilt.

### 4.1. The Contribution of Interpersonal Characteristics to PGD among Suicide-Loss Survivors

Our findings offer evidence that interpersonal factors can have a positive and protective effect on the development of PGS among SLSs. We suggest that sense of belonging, considered one of the basic psychological needs of humankind [60], may provide a significant protective barrier to these grief reactions and may thus help SLSs to better cope with different psychological difficulties [61], even with suicidal ideation and behavior [36,62]. Moreover, our results suggest that self-disclosure may play an essential role in coping with grief after the suicide of a loved one. Previous studies [33] found that encouraging survivors to self-disclose promotes the exchange of social support and the experience of belonging, connectedness, and togetherness, which may inhibit complications in grief [33]. Self-disclosure may help organize intrapsychic grief and process the emotional aspects of the traumatic suicide event [61,63], especially for individuals who have difficulty accepting and regulating their emotions [64,65]. When SLSs conceal the suicidal event or avoid talking about it (for example, because they perceive themselves as a burden to those around them), they risk shutting themselves off from people who are a potential source of help, comfort, and support [41]. Similarly, Oexle et al. [66] showed that the tendency to maintain a veil of secrecy regarding suicide loss is associated with more severe grief difficulties among SLSs.

These findings are not surprising considering that coping with loss after a suicide has been shown to have severe effects on relationships in the social support systems of SLSs [41,43]. Many SLSs feel that their relationships with those around them become more distant after the suicidal event [67], and they are less confident in and less reliant on their social networks [68]. Moreover, these findings are consistent with the new DSM-5-TR criteria, which indicate intense loneliness and a sense of detachment from others as symptoms (listed in Criterion C) for the diagnosis of PGD [5]. In addition, we may speculate that damage to the supportive interpersonal network of a SLS may exacerbate other grief reactions listed as symptoms in the DSM-5-TR. For example, SLSs may experience significant disruption in their social relationships due to a desire to stave off reminders of the deceased. These unwanted reminders and cues may include people, places, or situations that could trigger their sense of loss and other painful feelings related to the deceased and the suicide event (listed as a symptom in Criterion C for the diagnosis of PGD according to the DSM-5-TR criteria) [5]. The resultant profound loneliness may even intensify their longing for and preoccupation with the deceased person (listed as a symptom in Criterion B for the diagnosis of PGD according to the DSM-5-TR criteria) [5]. We recommend that these issues be investigated in future longitudinal studies to demonstrate clear causality.

Thus, experiences of belongingness and solidarity may help SLSs cope with painful feelings of grief, including social isolation which could lead to PGD, as formulated in the DSM-TR-5 [5]. Knowing that one can rely on the support of friends and family and not face a lonely future can help soften the blow of loss and possibly ward off its harmful effects. These benefits can be achieved, among others, through religious affiliation, which may provide a sense of belonging [69]. Religious affiliation is thought to help cope with grief; it has been shown to protect against painful emotions associated with grief [70].

### 4.2. The Moderating Role of Perceived Burdensomeness in the Association between Guilt and PGD among Suicide-Loss Survivors

Our results suggest that PB levels significantly moderate the contribution of guilt to PGS. PB is interactively related to interpersonal variables such as social belonging, social support, and self-disclosure [37]. Therefore, we suggest that SLSs who experience intense feelings of guilt following the suicide of a loved one but also feel justified remaining within their immediate social environment may incur significant benefits. These SLSs will then likely benefit from the healing qualities of a sense of belonging and togetherness and the opportunity to share the distressing guilt they carry in their hearts with those around them. Conversely, when SLSs who experience guilt are confronted with the belief that they are a burden to those around them, the emotional pain they carry can intensify and interfere with their grieving process in several ways.

Guilt may reinforce survivors’ perceptions that they are burdensome to those around them during the grieving process [22,57]. After the loss, many SLSs ruminate intensely about the conceivable reasons for the suicidal event and their role in it [56,71,72] and often exaggerate their responsibility [20,73]. In addition, the internal guilt that many SLSs feel regardless of their environment [67] may be met with society’s negative stereotypes and prejudices [43,74], which has recently been associated with higher levels of perceived burden [75]. Thus, guilt may reinforce SLSs’ perception that they are so defective that their presence could harm and burden their loved ones or society in general [75].

SLSs may also feel that the expression of guilt, loneliness, and grief burdens their social environment, particularly if their acquaintances are grieving for the same person who died of suicide [59]. Thus, they may be reluctant to share their feelings of guilt with others, preferring to remain alone with them [76]. This perception may even lead SLSs to avoid entering new relationships for fear of further loss because they believe they are “cursed” [67]. Therefore, it is likely that this interactive relationship may increase feelings of loneliness and disconnection from their environment. In addition, we suggest that SLSs’ deeply held belief that their loved ones would be better off without them exacerbates their feelings of guilt. For example, when SLSs experience themselves as a burden, they inevitably feel dependent on others to some degree and may experience frustration and guilt for the resulting difficulties [77]. Finally, PB and guilt contain a strong emotional component associated with self-loathing [36,78], and their coexistence could dramatically exacerbate their inner turmoil, self-accusation, and emotional distress following the loss. This perspective would be interesting to explore in future studies.

Thus, we suggest that the destructive relationship between guilt and feelings of burdensomeness may interfere with normative grief processes and increase the likelihood of PGD among SLSs. This process may be driven by severely limiting the individual’s capacity to seek help in their immediate environment, deepening feelings of isolation, being cut off from others, and increasing self-hatred and emotional distress. In addition, this interactive relationship can interfere with the ability of the SLS to experience hope and vitality, making their future seem bleak, empty, and meaningless. These perceptions and feelings correspond to the symptoms listed in Criterion C for the diagnosis of PGD, as articulated in the DSM-TR-5 [5].

### 4.3. Limitations and Future Directions

This study has several methodological limitations. The first limitation concerns the voluntary nature of the sample. Most participants were members of organizations and forums concerned with loss in the wake of suicide. Therefore, individuals with some potential for enduring pain may be overrepresented in the current sample. In addition, the recruitment process likely yielded an overrepresentation of people active on the Internet and social media. Consequently, our sample may comprise fewer individuals with high levels of distress who do not have Internet access or are not active on social media platforms.

Furthermore, our sample consisted mainly of participants with a high education level and a high proportion were women (80.4%), limiting the generalizability of the current results to other populations. However, even in representative samples of SLSs, women are typically overrepresented to some degree, given that men are three times more likely than women to die of suicide [58]. However, as women are often more prone to higher levels of guilt [79], gender should be considered in future studies. In addition, the overrepresentation of secular individuals in our sample (73.7%) may bias the study findings due to the protective effects of religion and spirituality on grief [70] and guilt [80]. Finally, using an Israeli sample with specific cultural scenarios regarding grief and sharing [81] may limit the generalizability of our findings to other cultures. Therefore, a broader and more representative sample of SLSs will facilitate a more comprehensive and representative understanding of PGS among suicide survivors.

Another limitation relates to the type of questionnaires used in our study. The use of self-report questionnaires may introduce various biases in participants’ reports (e.g., social desirability), a factor particularly relevant when addressing the sensitive topic of PGD and guilt following a suicide. Furthermore, PGD identification was derived from self-report questionnaires derived from cross-sectional samples. Future studies should therefore consider collecting data from multiple sources of information and incorporate more objective measures. For example, a more rigorous method for identifying PGD could include systematic clinical interviews where trained mental health professionals would make the diagnosis.

Finally, the study’s cross-sectional nature and the lack of an assessment of interpersonal functioning before the suicidal event limit our ability to determine the order of the phenomena and thus the causality of these relationships. The effects of guilt, PB, TB, and SD on PGS may be interactive and circular. Moreover, some participants may have had a history of mental disorders (e.g., depression) and may have been prone to guilt prior to the suicidal event [82]. Therefore, we cannot rule out the possibility that a participant’s current report of guilt and depression reflects circumstances that preceded the suicidal event and contributed to its affect while not being directly related to the suicide. Thus, we recommend that future studies consider the psychiatric history of participants and conduct a longitudinal study that might reveal the approximate causal relationship between intrapersonal and interpersonal characteristics and PGS.

### 4.4. Theoretical and Practical Implications

Despite the noted limitations, the present study has theoretical and practical implications. Concerning theoretical implications, the study’s findings highlight PB’s crucial and moderating role in the relationship between guilt and PGD among SLSs. Thus, our findings suggest that the recovery process relies upon SLSs’ interpersonal experiences and perceptions when dealing with the suicide of a loved one, especially for those with high levels of guilt.

Feelings of guilt and other painful emotions associated with death may impact SLSs’ social networks by creating the harsh perception that they are a burden to those around them. These sensations and perceptions interfere with normative grieving processes, impede receiving support and sharing their pain, and increase the likelihood that they will suffer from PGD. Thus, the study findings highlight the importance of the resilience factors of sense of belonging and self-disclosure, which can be served as barriers to PGS among SLSs. It is noteworthy that the contribution of guilt and perceived burdensomeness on PGS is likely interactive and circular, as high levels of PGS also contribute to increases in guilt and PB which, in turn, increase the severity of PGS. Therefore, it would be essential to investigate this complex relationship in future longitudinal studies, which could contribute to designing targeted interventions for SLSs.

Concerning practical implications, our findings suggest that interpersonal variables may serve as buffers against grief complications, particularly among SLSs who suffer from painful guilt. Therefore, health care professionals and the general environment surrounding the SLS should monitor the social involvement of the SLS and the extent of social support they receive. Furthermore, disclosing intimate feelings and thoughts may protect against PGD by enhancing a sense of belonging, relieving harmful cognitions (e.g., perceived burdensomeness), and reducing guilt [23,83].

It is noteworthy that PB and TB typically reflect distorted perceptions and are not based on facts [84]. Therefore, primary psychoeducational interventions could help SLSs gain support and a new perspective on themselves and the suicide event [85]. In addition, psychotherapy protocols aiming to help people understand their relationships with others may benefit individuals with high TB and PB levels. For example, interpersonal psychotherapy (IPT), which aims to improve interpersonal skills, especially self-disclosure, may prove effective for SLSs [86]. Moreover, psychosocial therapies focusing on activating social support (to reduce feelings of thwarted belongingness) and reducing maladaptive cognitions (i.e., helping survivors find ways to contemplate death that do not evoke strong feelings of guilt or perceived burdensomeness) may benefit SLSs [1].

Another possible treatment avenue may be group therapy, which can facilitate SLSs sharing and disclosing their feelings of guilt, modifying the meaning of the suicide event, normalizing the grief experience of suicide, and, most importantly, providing them with a sense of belonging [87]. However, peer support programs may provide similar benefits [88], where SLSs can share their feelings and personal stories with each other, validate their grief experience, and exchange psychosocial support that strengthens their sense of belonging and “togetherness” [88]. Furthermore, our findings point to the potential of mindfulness-based interventions to promote self-compassion, which has been shown to positively impact feelings such as TB and PB [32,65], as well as coping with unpleasant emotions such as guilt [89]. These approaches could help the bereaved overcome the painful loss of a loved one and even exhibit growth in the wake of these painful circumstances [32].

## 5. Conclusions

In conclusion, this study has expanded the current knowledge of the factors and mechanisms contributing to PGD according to the criteria proposed in the DSM-5-TR [3] after a suicide loss. Moreover, the study’s findings indicate the crucial and moderating role of perceived burdensomeness in the relationship between guilt and PGD among SLSs. It can be suggested that guilt and perceived burdensomeness may have a detrimental interaction on a person’s connection to their social network, thus interfering with normative grief processes and increasing the risk of PGD among SLSs. In addition, the study’s findings highlight the importance of social and community factors in preventing PGD among suicide-loss survivors and suggest that the resilience factors of sense of belonging and self-disclosure may help SLSs to effectively cope with PGS.

## Figures and Tables

**Figure 1 ijerph-19-10545-f001:**
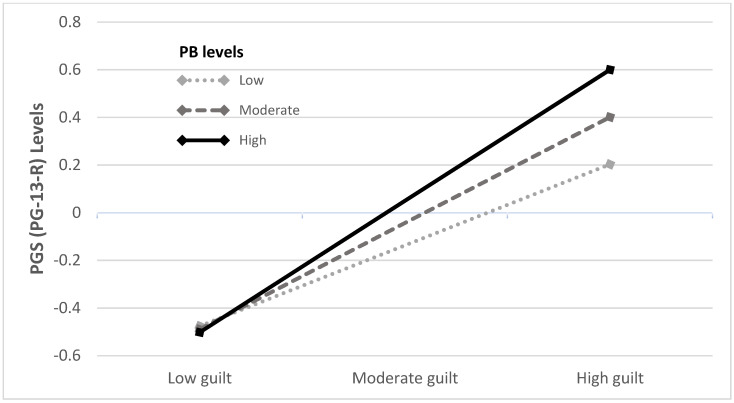
The association between guilt and PGS as moderated by PB levels (*n* = 152). PB = perceived burdensomeness. PGS = prolonged grief symptoms.

**Table 1 ijerph-19-10545-t001:** Means, standard deviations, and intercorrelations between main study measures (*n* = 152).

	Measures	1	2	3	4	5	6	7	8	9
1	Time since suicide ^1^	1								
2	Participant’s age at the time of suicide	−0.41 **	1							
3	Fear of a future suicide of a family member	−0.17	−0.12	1						
4	Depression	−0.08	0.13	0.42 ***	1					
5	Guilt	−0.09	0.13	0.34 ***	0.59 ***	1				
6	PB	−0.09	−0.09	0.47 ***	0.55 ***	0.42 ***	1			
7	TB	−0.18	0.14	0.25 ***	0.42 ***	0.33 ***	0.71 ***	1		
8	Self-disclosure	0.12	−0.31 ***	−0.07	−0.18 *	−0.22 **	−0.24 **	−43 **	1	
9	PGS	−0.18	0.32 ***	0.35 ***	0.71 ***	0.57 ***	0.37 ***	0.19 *	−0.30 ***	1
	*Mean*	166.03	30.68	4.84	6.41	3.53	12.90	20.35	3.66	2.08
	*Standard Deviation*	105.52	10.13	1.14	3.61	1.03	3.82	5.39	0.73	0.50
	Range: Min-max	24–672	9–60	1–7	0–25	1–10	8–33	9–42	1–7	1–4

* *p* < 0.05, ** *p* < 0.01, *** *p* < 0.001. PB = perceived burdensomeness; TB = thwarted belongingness; PGS = prolonged grief symptoms according to the PG-13-R scale. ^1^ In months.

**Table 2 ijerph-19-10545-t002:** Summary of hierarchical multiple regression analysis for predicting PGS (*n* = 152).

Predictors	B	SE B	β	∆R^2^	R^2^/Adj. R^2^	*F* _change_
**Step 1**				0.25 ***	0.25/0.23	16.08 ***
Time since suicide ^1^	0.00	0.01	−0.02			
Participant’s age at the time of suicide.	0.34	0.07	0.37 ***			
Fear of a future suicide of a family member	0.34	0.06	0.38 ***			
**Step 2**				0.24 ***	0.49/0.47	34.95 ***
Depression	0.17	0.07	0.45 ***			
Guilt	0.44	0.06	0.19 **			
**Step 3**				0.06 ***	0.55/0.52	5.79 ***
PB	0.16	0.08	0.16			
TB	0.29	0.08	0.32 ***			
Self-disclosure	−0.19	0.06	−0.21 **			
**Step 4**				0.05 ***	0.60/0.57	6.15 ***
Guilt X PB	0.21	0.05	0.46 ***			
Guilt X TB	0.12	0.05	0.30 *			
Guilt X SD	−0.05	0.06	−0.06			

Degrees of freedom for Step 1, *F*(3, 148); for Step 2, *F*(5, 146); for Step 3, *F*(8, 143); and for Step 4, *F*(11, 140). * *p* < 0.05. ** *p* < 0.01. *** *p* < 0.001. PB = perceived burdensomeness; TB = thwarted belongingness; SD = self-disclosure; PGS = prolonged grief symptoms as measured by the PG-13-R scale. ^1^ no. months.

## Data Availability

Due to ethical concerns, supporting data cannot be made openly available. Further information about the data and conditions for access can be received by approaching the corresponding author.

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
