# Peer review of "Prolonged Grief Symptoms among Suicide-Loss Survivors: The Contribution of Intrapersonal and Interpersonal Characteristics"

_ijerph, 2022, doi:10.3390/ijerph191710545_

Round 1

Reviewer 1 Report

I thank the authors for the substantial modifications made to the first manuscript.

The modifications made are pertinent.

Only two minor details remain. In table 2 it is necessary to include the meaning of the abbreviation “SD” that appears in the last line of the table. In figure 1 it is necessary to include the meaning of the abbreviation "PGS"

Author Response

Only two minor details remain. In table 2 it is necessary to include the meaning of the abbreviation “SD” that appears in the last line of the table. In figure 1 it is necessary to include the meaning of the abbreviation "PGS"

Thank you. as suggested, we changed that in the revised version.

Reviewer 2 Report

Thank you for addressing the previous comments so thoroughly. I hope you feel the presentation of this valuable work has been improved.

A few minor comments:

Thank you for adding further clarification of religiosity on lines 268-269. Please consider if the breakdown of all categories is needed and the appropriateness of presenting the religious affiliation of one participant alone.

Line 409 ‘These benefits can be achieved through religious affiliation, which may provide a sense of belonging’. Add clarity that religion may be one factor that can increases a sense of belongingness, among others.

Throughout the discussion, implications and conclusion, it would be beneficial if the authors could ensure their language differentiates between findings of the study, that can be articulated with more definitive language, and the hypothesised mechanisms to explain findings, which are not supported directly by the study and require further research and thus more tentative language. For example, line 448, it is stated ‘this process is driven by’ where it would seem the language should be more tentative.

In a number of places (e.g. line 375, 552) , it is discussed how a sense of belonging and self-disclosure may ‘shield’ SLSs from PGS. That language seems very strong, given that the study has identified other predictors that explain much of the variance in PGS. Please consider revising these terms.

Author Response

Thank you for addressing the previous comments so thoroughly. I hope you feel the presentation of this valuable work has been improved.

Thank you for your inputs!

A few minor comments:

Thank you for adding further clarification of religiosity on lines 268-269. Please consider if the breakdown of all categories is needed and the appropriateness of presenting the religious affiliation of one participant alone.

Thank you, as suggested we changed that in the revised version.

Line 409 ‘These benefits can be achieved through religious affiliation, which may provide a sense of belonging’. Add clarity that religion may be one factor that can increases a sense of belongingness, among others.

Thank you, as suggested we changed that in the revised version.

Throughout the discussion, implications and conclusion, it would be beneficial if the authors could ensure their language differentiates between findings of the study, that can be articulated with more definitive language, and the hypothesised mechanisms to explain findings, which are not supported directly by the study and require further research and thus more tentative language. For example, line 448, it is stated ‘this process is driven by’ where it would seem the language should be more tentative.

Thank you, as suggested we changed that in the revised version and tempered the tone in several places in the discussion.

In a number of places (e.g. line 375, 552) , it is discussed how a sense of belonging and self-disclosure may ‘shield’ SLSs from PGS. That language seems very strong, given that the study has identified other predictors that explain much of the variance in PGS. Please consider revising these terms.

Thank you, as suggested we tempered the tone in these sentences.

Reviewer 3 Report

The revision addressed my suggestions. From my point of view the paper is ready for publication. 

Author Response

The revision addressed my suggestions. From my point of view the paper is ready for publication. 

Thank you for your inputs!

This manuscript is a resubmission of an earlier submission. The following is a list of the peer review reports and author responses from that submission.

Round 1

Reviewer 1 Report

This is an interesting manuscript on prolonging grief in suicide survivors.

The abstract is well crafted and agrees with the results.

The introduction is adequate for the purpose of the manuscript.

The method is clearly described.

The results are clearly presented except for a typographical error and lack of clarity in the tables and figures.

It is important that the tables and figures can stand alone, so it is necessary not to include abbreviations or include the meaning of the abbreviations as the foot of the table or figure.

On line 307 there seems to be a typo. It is noted that TB has a positive contribution to PGS when the B value is negative (β = -.32, t(143) = -3.80 …

The discussion and conclusions do not fully adhere to the results.

The results mention that only PB is a moderating variable of guilt and depression among the participants. TB has no moderating effect on guilt and depression. However, in the discussion and conclusions it is pointed out that both variables PB and TB have a moderating effect on guilt and depression. It is necessary to correct this lack of coherence between the results, the discussion, and the conclusions. Furthermore, it is necessary to further discuss why PB only had a moderating effect on depression and guilt.

In addition, it is necessary to discuss the low contribution of TB, PB and SD in the prediction of PGS observed in the multiple regression.

In the limitations, it is necessary to point out the overrepresentation of women and secular participants and its possible effects on the findings.

Reviewer 2 Report

Thanks for the opportunity to review this interesting and important paper.

I think it is a well-argued paper identifying an important gap in the current knowledge. Even though there are some references to the author themselves I think they are well-chosen and relevant for the paper. The measures used are described and explained.  

A disclaimer is that I am not an expert in the methods used, but the design seems appropriate to test the hypothesis. I think the manuscript are clear and well structured. 

I only have a few suggestions for the introduction:

Line 72: You write: Suicide is a significant public health problem, causing an estimated 800,000 deaths 71 annually, making it the seventeenth leading cause of death worldwide [9].

I would suggest using a newer reference: World Health Organisation. Suicide worldwide in 2019: global health estimates. Geneva: World Health Organization; 2021; Licence: CC BY-NC-SA 3.0 IGO (9789240026643-eng (1).pdf) where the number of death by suicide is 703 000 people every year.

Line 78 – 81: I think it is possible to find newer references supporting your statements in these lines.

Line 103 – 104: You mentioned several studies, but only have one example – I would like more examples to underline the existence of “several studies”

Reviewer 3 Report

Well done to the authors on examining an important area of prolonged grief following suicide loss and the impact of this on a person’s wellbeing. By identifying factors that contribute to prolonged grief and how they interact, this work is helping to identify tangible areas to focus support and therapy for suicide loss survivors to reduce the impact of prolonged grief symptoms. There are areas where I feel the manuscript can be improved and hope these comments help to progress the manuscript:

·       There are many acronyms within the abstract and I believe the ease of reading may be improved if the number were reduced within the abstract. Similarly, acronyms make parts of the discussion challenging to follow.

·       It would be useful if examples of both the intrapersonal and interpersonal factors were given where initially mentioned in the abstract.

·       The introduction is quite lengthy and I would encourage the authors to consider if it can be condensed further for readers. Perhaps a table could be used to explain the many concepts that are defined in the introduction or a figure to describe the hypothesized relationships.

·       Line 72-74: Perhaps this statement could be rephrased to capture that 135 people are exposed to each suicide and many of these may be adversely affected.

·       A more explicit link could be made in applying the ‘interpersonal theory of suicide’ to suicide loss.

·       The first statement of the methods highlights this study is part of a longitudinal study but I understand that the data for this study are cross-sectional i.e. just one timepoint? This should be clearer at the outset of the methods and in the abstract.

·       While the methods refer to a previous study, some additional succinct summary details would be helpful to include in this manuscript e.g. including the country where participants were based and process of recruitment.

·       The results section is well structured and clear.

·       I suggest considering correction of statistical significance for running multiple analyses.

·       Clarify terminology on line 257: ‘being traditional’.

·       In the discussion section, I feel at times this is a very broad review of literature (Section 4.1) and it needs to be focused more on explicitly explaining the findings from this study and how they add to existing knowledge. For example, self-disclosure and thwarted belongingness contributed directly to prolonged grief symptoms. The initial interaction analysis did not support an interaction between self-disclosure and guilt and the later moderation analysis found a significant moderation of guilt by perceived burdensomeness but not thwarted belongingness. It is not clear to me that the discussion addresses these nuances.

·       Depression is mentioned in the introduction as an outcome of suicide loss. The rationale for including guilt as an intrapersonal factor is clear. Greater justification for including depression over other factors would be useful. In the discussion it is stated that ‘Our results indicated that guilt and depression are critical contributors to PGS development beyond sociodemographic and loss-related factors’ (Line 340-1). However, if I understand correctly, these concepts are measured cross-sectionally, so this statement seems to go beyond the study findings. Circularity is noted in the implications section. This limitation should also be clear when discussing the cross-sectional nature of the research in the limitations. Was there any potential to examine the relationship prospectively in the longitudinal study?

·       Line 398-400: This does not seem to be a finding of the current study. Perhaps the authors could clarify if this idea is supported by findings from the current study in conjunction with other research.

·       Within the strengths and limitations, it would be helpful to consider any role that compensation may have had in influencing the characteristics of the sample.

·       Line 471-4: consider revising/simplifying this sentence to clarify its meaning.

·       Line 498-501: The potential for group therapy to address these interpersonal factors seems reasonable and perhaps some mention could also be made to peer support which could encompass many of the desirable mechanisms listed.

·       Line 511-4: the phrasing of this sentence suggests a mediation relationship whereas the focus of the tests in the analysis have been on moderation.

·       Please include a statement on data availability.

I hope these points are helpful for the authors.